# Solution-Processable Colorless Polyimides Derived from Hydrogenated Pyromellitic Dianhydride: Strategies to Reduce the Coefficients of Thermal Expansion by Maximizing the Spontaneous Chain Orientation Behavior during Solution Casting

**DOI:** 10.3390/polym14061131

**Published:** 2022-03-11

**Authors:** Masatoshi Hasegawa, Katsuki Ichikawa, Shuichi Takahashi, Junichi Ishii

**Affiliations:** Department of Chemistry, Faculty of Science, Toho University, 2-2-1 Miyama, Funabashi, Chiba 274-8510, Japan; www.ichikawa@gmail.com (K.I.); ufsbopdi8-35127-74-2@ezweb.ne.jp (S.T.); junishii@sci.toho-u.ac.jp (J.I.)

**Keywords:** polyimides, hydrogenated pyromellitic dianhydride (H-PMDA), optical transparency, coefficients of thermal expansion (CTE), toughness, solution processability, surface hardness, heat-resistant plastic substrates

## Abstract

In this study, practically useful colorless polyimides (PIs) with low coefficients of thermal expansion (CTEs) and other desirable properties were prepared from hydrogenated pyromellitic dianhydride (1-*exo*,2-*exo*,4-*exo*,5-*exo*-cyclohexanetetracarboxylic dianhydride, H-PMDA). A modified one-pot polymerization method afforded a high-molecular-weight PI with sufficient film-forming ability from 2,2′-bis(trifluoromethyl)benzidine (TFMB) with a rod-like structure and H-PMDA. However, the PI film cast from its homogeneous solution did not have low CTEs, similar to the analogous system using *meta*-tolidine. To solve this problem, a series of amide- and amide-imide-containing diamines were designed and synthesized. The modified one-pot polymerization of H-PMDA and the diamines in *γ*-butyrolactone produced homogeneous, viscous, and stable solutions of high-molecular-weight PIs with high solid contents. The cast films of certain systems examined in this study simultaneously achieved low CTEs, high optical transparency, considerably high glass transition temperatures (*T*_g_s), and sufficient ductility. A possible mechanism for the generation of low CTEs, which is closely related to the spontaneous in-plane orientation behavior during solution casting, was proposed. Certain H-PMDA-based PIs developed in this study are promising colorless heat-resistant plastic substrates for use in image display devices and other optical applications.

## 1. Introduction

Wholly aromatic polyimides (PIs) have been used as the most reliable electrically insulating materials in various electronic devices because of their excellent combined properties, including their considerably high physical and chemical heat resistances (e.g., resistance to the soldering process at 260 °C), good mechanical properties, high flame retardancy, extremely high purity (absence of the residues of monomers and solvents, as well as ionic, metallic, and halogen impurities), resistance to various chemicals, and high dimensional stability [1,2,3,4,5,6,7,8,9,10,11]. However, the intense coloration of wholly aromatic PI films, which arises from intra- and intermolecular charge–transfer (CT) interactions [12], often disturbs their optical and optoelectronic applications (e.g., their use as plastic substrates and as alternatives to current non-alkali glass substrates in image display devices). Optically transparent (colorless) PIs are promising heat-resistant plastic substrate materials. Replacement of the currently used glass substrates (thickness ~400 μm) with plastic substrates (thickness < 50 μm) renders the resultant display devices considerably lighter, thinner, and more flexible (ultimately foldable).

The most effective way to completely remove the coloration of PI films is to use aliphatic (usually, cycloaliphatic) monomers either in diamines, tetracarboxylic dianhydrides, or both [12,13,14,15,16,17,18,19,20,21,22,23,24,25,26,27,28,29,30,31,32,33,34,35,36,37,38], thereby preventing CT interactions. However, when aliphatic diamines are used, poorly soluble salts are formed in anhydrous amide solvents in the early stage of polyaddition [30], which hampers the formation of the PI precursors, viz. poly(amic acid)s (PAAs) with high molecular weights, or requires extremely long times for the reaction mixture to become homogeneous [30,39,40]. Thus, salt formation is a significant obstacle to the industrial production of aliphatic-diamine-based colorless PIs because of the poor productivity and reproducibility during polyaddition. In contrast, no salt is formed in PI systems obtained from cycloaliphatic tetracarboxylic dianhydrides and aromatic diamines [18,19,20,21,22,23,24,25,26,27,28,29,30,31,32,33,34]. However, currently, industrially produced cycloaliphatic tetracarboxylic dianhydrides with acceptable manufacturing costs and performance are virtually limited to hydrogenated pyromellitic dianhydride (1-*exo*,2-*exo*,4-*exo*,5-*exo*-cyclohexanetetracarboxylic dianhydride, H-PMDA [41]) and *anti*-1,2,3,4-cyclobutanetetracarboxylic dianhydride (CBDA) [18], indicating less scope for structural modifications. In addition, it is difficult to significantly reduce the cost of CBDA because its synthetic process is based on ultraviolet (UV) light irradiation, which is not suitable for mass production, unlike common thermal reactions. Therefore, the combination of H-PMDA and aromatic diamines is the only choice for the industrial production of colorless PIs.

Plastic substrates are required to exhibit non-coloration, extremely high glass transition temperatures (*T*_g_s, desirably > 350 °C), and sufficient film ductility, as well as low coefficients of thermal expansion (CTEs, desirably < 20 ppm K^−1^) in the film plane (*X*–*Y* direction) in the glassy temperature regions, in order to ensure excellent dimensional stability during multiple heating–cooling cycles in device fabrication processes. The significant thermal expansion–contraction behavior of the plastic substrates during the thermal cycles can lead to serious problems, including misalignment and adhesion failure of various micro-components, laminate warpage, and transparent electrode breakdown. However, most polymeric materials do not exhibit low-CTE characteristics unless the films are highly stretched, as suggested by their high CTE values (60–120 ppm K^−1^) [42]. For example, a poly(ether sulfone) film cast from *N*-methyl-2-pyrrolidone (NMP), as well as NMP-cast films of highly soluble PIs, showed a common CTE (65 ppm K^−1^) with practically zero birefringence in the *Z*-direction (Δ*n*_th_ = 0.0001), suggesting that solution casting does not usually induce low CTEs.

Exceptionally, PIs comprising rigid and linear backbones afford low-CTE films by prominently aligning the main chains in the *X***–***Y* direction (in-plane orientation) during the thermal imidization of PAA films fixed on substrates (conventional two-step process) [43,44,45,46]. However, the H-PMDA-based thermally imidized films did not show low CTEs [23]. This suggests that the non-linear and non-planar steric structure of the H-PMDA-based diimide units significantly reduced the overall main-chain linearity, and consequently the imidization-induced in-plane orientation was disturbed.

We recently found that simple solution casting of specific PI systems induces lower CTEs and higher optical transparency than the counterparts prepared via the conventional two-step process [24,30,47,48,49,50]. This effect (casting-induced spontaneous in-plane orientation) became more prominent with increasing molecular weight of the PIs [24]. Nevertheless, the casting of H-PMDA-based PI solutions was ineffective in reducing the CTE. Thus, there was no way to achieve a low CTE as long as H-PMDA was used. 

In this study, we aimed to obtain practically useful H-PMDA-based colorless PI films with low CTEs and other desired properties (considerably high heat resistance and good toughness) by combining the use of novel amide-linked diamines and the modification of the one-pot polymerization process, which can maximize the spontaneous chain orientation behavior induced during solution casting [51].

## 2. Experimental Section

### 2.1. Materials

#### 2.1.1. Monomer Synthesis

##### Bisamide-Type Diamines

In this study, a series of bisamide-type diamines was synthesized according to the reaction schemes shown in Figure 1. As an example of the bisamide-type diamines, the detailed synthetic procedures of AB-44ODA (Figure 1) are described below.

AB-44ODA: In a sealed flask, 4,4′-oxydianiline (4,4′-ODA, 15 mmol) was dissolved in anhydrous tetrahydrofuran (THF, 15 mL) in the presence of pyridine (Py, 75 mmol, 6 mL) as an HCl acceptor. 4-Nitrobenzoyl chloride (4-NBC, 34 mmol) was dissolved in anhydrous THF (15 mL) in another sealed flask with a septum cap. To the 4,4′-ODA solution cooled at 0 °C, the 4-NBC solution was slowly added using a syringe with continuous magnetic stirring, after which the reaction mixture was stirred at 0 °C for several hours, then subsequently at room temperature for 12 h. The precipitate formed was collected by filtration, washed with toluene (100 mL) and a large quantity of water, and dried at 120 °C under vacuum for 12 h (yield: 90%). The pale yellowish product showed a sharp endothermic peak for melting at 272 °C by differential scanning calorimetry (DSC, Netzsch, DSC3100) conducted at a heating rate of 5 °C min^−1^. The molecular structure of the product was confirmed to be the desired nitro compound (NB-44ODA) from the FT-IR (Jasco, FT/IR-4100) and ^1^H-NMR (JEOL, JNM-ECP400) spectra. FT-IR (KBr plate method, cm^−1^): 3357 (amide N–H), 3109 (C_arom_–H), 1654 (amide-I, C=O), 1541 (NO_2_ + amide-II, C=O), 1500 (1,4-phenylene), 1253 (C_arom_–O–C_arom_). ^1^H-NMR [400 MHz, dimethyl sulfoxide (DMSO)-*d*_6_, *δ*, ppm]: 10.60 [s, 2H (relative integrated intensity: 2.00H), NHCO], 8.38 [dd, 4H (3.97H), *J* = 9.0, 2.0 Hz, 2,2′,6,6′-protons of the terminal nitrobenzene units (NB)], 8.20 [dd, 4H (4.10H), *J* = 8.9, 2.0 Hz, 3,3′,5,5′-protons of NB], 7.81 [dd, 4H (4.09H), *J* = 9.1, 2.1 Hz, 3,3′,5,5′-protons of the central diphenyl ether unit (DE)], 7.06 [dd, 4H (4.11H), *J* = 9.0, 2.2 Hz, 2,2′,6,6′-protons of DE]. 

The catalytic reduction of NB-44ODA was conducted while tracking the reaction with thin-layer chromatography as follows. NB-44ODA (13.2 mmol) was dissolved in *N*,*N*-dimethylacetamide (DMAc, 30 mL) in the presence of Pd/C (0.70 g) as a catalyst. The reaction mixture was refluxed at 80 °C for 4 h in a hydrogen atmosphere and cooled to room temperature without precipitation of the product. After the catalyst residue was removed by filtration, the filtrate was slowly poured into a large quantity of water. The white precipitate formed was washed with acetone and dried at 80 °C for 12 h under vacuum (yield: 92%). The molecular structure of the product was confirmed to be the desired diamine compound (AB-44ODA) from the following analytical data. Melting point (DSC): 282 °C. FT-IR (KBr plate method, cm^−1^): 3409 (amine N–H), 3375/3322 (amine + amide N–H), 3036 (C_arom_–H), 1658 (amide C=O), 1509 (1,4-phenylene), 1257 (C_arom_–O–C_arom_). ^1^H-NMR (400 MHz, DMSO-*d*_6_, *δ*, ppm): 9.78 [s, 2H (2.00H), NHCO], 7.76−7.71 [m, 8H (8.04H), 3,3′,5,5′-protons of the terminal aniline unit (AN) + 3,3′,5,5′-protons of DE], 6.97 [d, 4H (4.10H), *J* = 9.1 Hz, 2,2′,6,6′-protons of DE], 6.61 [d, 4H (4.09H), *J* = 8.6 Hz, 2,2′,6,6′-protons of AN], 5.75 [s, 4H (4.02H), NH_2_]. Elemental analysis, Anal. Calcd (%) for C_26_H_22_O_3_N_4_ (438.49): C, 71.22; H, 5.06; N, 12.78. Found: C, 71.21; H, 5.16; N, 12.64. 

Other bisamide-type diamines were synthesized in a similar manner. Their analytical data are shown below.

AB-34ODA: Melting point (DSC): 244 °C. FT-IR (KBr plate method, cm^−1^): 3483/3433/3403/3230 (amine N–H), 3356 (amine + amide N–H), 3043 (C_arom_−H), 1631 (amide C=O), 1503 (1,4-phenylene), 1264 (C_arom_–O–C_arom_). ^1^H-NMR (400 MHz, DMSO-*d*_6_, *δ*, ppm): 9.81, 9.81 [s + s, 2H (2.00H), NHCO], 7.79 [d, 2H (2.02H), *J* = 8.0 Hz, 3,5-protons of DE], 7.74–7.68 [m, 4H (3.95H), 3,5- + 3′,5′-protons of AN], 7.54–7.49 [m, 2H (1.98H), 2′- + 4′-protons of DE], 7.29 [t, 1H (0.99H), *J* = 8.0 Hz, 5′-proton of DE], 7.03 [d, 2H (2.01H), *J* = 8.0 Hz, 2,6-protons of DE], 6.68 [dd, 1H (0.98H), *J* = 8.0, 2.3 Hz, 6′-proton of DE], 6.69–6.57 [m, 4H (4.01H), 2,6- + 2′,6′-protons of AN], 5.76, 5.75 [s + s, 4H (4.00H), NH_2_]. Elemental analysis, Anal. Calcd (%) for C_26_H_22_O_3_N_4_ (438.49): C, 71.22; H, 5.06; N, 12.78. Found: C, 71.21; H, 5.26; N, 12.62.

AB-*m*TOL: Melting point (DSC): 295 °C. FT-IR (KBr plate method, cm^−1^): 3458 (amine N–H), 3376/3305 (amine + amide N−H), 3037 (C_arom_−H), 2917 (C_aliph_−H), 1650/1530 (amide C=O), 1508 (1,4-phenylene). ^1^H-NMR (400 MHz, DMSO-*d*_6_, *δ*, ppm): 9.75 [s, 2H (2.00H), NHCO], 7.74 [d, 4H (4.03H), *J* = 8.6 Hz, 3,3′,5,5′-protons of AN], 7.69 [sd, 2H (1.93H), *J* = 1.9 Hz, 3,3′-protons of the central biphenylene unit (BP)], 7.61 [dd, 2H (1.99H), *J* = 8.3, 2.0 Hz, 5,5′-protons of BP], 7.00 [d, 2H (1.96H), *J* = 8.2 Hz, 6,6′-protons of BP], 6.61 [d, 4H (4.02H), *J* = 8.6 Hz, 2,2′,6,6′-protons of AN), 5.75 [s, 4H (4.10H), NH_2_], 2.01 [s, 6H (6.03H), CH_3_]. Elemental analysis, Anal. Calcd (%) for C_28_H_26_O_2_N_4_ (450.54): C, 74.65; H, 5.82; N, 12.44. Found: C, 74.44; H, 5.99; N, 12.38.

AB-TFMB: Melting point (DSC): 317 °C. FT-IR (KBr plate method, cm^−1^): 3512/3418 (amine N–H), 3303 (amine + amide N−H), 3096/3039 (C_arom_−H), 1655 (amide C=O), 1509 (1,4-phenylene), 1311 (C−F). ^1^H-NMR (400 MHz, DMSO-*d*_6_, *δ*, ppm): 10.16 [s, 2H (2.00H), NHCO], 8.33 [sd, 2H (2.02H), *J* = 1.7 Hz, 3,3′-protons of BP], 8.07 [dd, 2H (1.97H), *J* = 8.4, 1.6 Hz, 5,5′-protons of BP], 7.77 [d, 4H (4.05H), *J* = 8.6 Hz, 3,3′,5,5′-protons of AN], 7.32 [d, 2H (2.03H), *J* = 8.4 Hz, 6,6′-protons of BP], 6.64 [d, 4H (4.02H), *J* = 8.5 Hz, 2,2′,6,6′-protons of AN], 5.86 [s, 4H (3.99H), NH_2_]. Elemental analysis, Anal. Calcd (%) for C_28_H_20_O_2_N_4_F_6_ (558.47): C, 60.22; H, 3.61; N, 10.03. Found: C, 60.00; H, 3.87; N, 9.98. 

AB-APAB: Melting point (DSC): 342 °C. FT-IR (KBr plate method, cm^−1^): 3334 (amine + amide N–H), 3034 (C_arom_–H), 1718 (ester C=O), 1655 (amide C=O), 1515 (1,4-phenylene). ^1^H-NMR (400 MHz, DMSO-*d*_6_, *δ*, ppm): 10.27 [s, 1H (1.00H), (C=O)-Ar-NHCO], 9.96 [s, 1H (1.05H), O-Ar-NHCO], 8.10−8.04 [m, 4H (3.99H), 2,6- + 3,5-protons of the benzoic acid (BA) unit in the central phenyl benzoate unit (PB)], 7.87–7.76 [m, 6H (6.17H), 3,5-protons of the phenol unit in PB + 3,5- + 3′,5′-protons of AN], 7.21 [d, 2H (1.99H), *J* = 8.0 Hz, 2,6-protons of the phenol unit in PB], 6.64–6.61 [m, 4H (4.08H), 2,6- + 2′,6′-protons of AN], 5.87 [s (br), 4H (3.98H), NH_2_].

AB-ATAB: Melting point (DSC): 317 °C. FT-IR (KBr plate method, cm^−1^): 3455 (amine N–H), 3315 (amine + amide N–H), 3039 (C_arom_–H), 2916 (C_aliph_–H), 1716 (ester C=O), 1650 (amide C=O), 1510 (1,4-phenylene). ^1^H-NMR (400 MHz, DMSO-*d*_6_, *δ*, ppm): 10.17 [s, 1H (1.00H), (C=O)-Ar-NHCO], 9.79 [s, 1H (1.02H), O-Ar-NHCO], 8.12 [d, 2H (1.96H), *J* = 8.9 Hz, 2,6-protons of the BA unit in PB], 8.02 [d, 2H (1.96H), 3,5-protons of the BA unit in PB], 7.78–7.74 [m, 5H (5.09H), 3-proton of the 2-methyphenol unit in PB + 3,5- + 3′,5′-protons of AN], 7.64 [dd, 1H (0.99H), *J* = 8.7, 2.5 Hz, 5-proton of the 2-methyphenol unit in PB], 7.13 [d, 1H (1.05H), *J* = 8.8 Hz, 6-proton of the 2-methyphenol unit in PB], 6.64–6.60 [m, 4H (4.05H), 2,6- + 2′,6′-protons of AN], 5.86 [s, 2H (2.00H), NH_a2_], 5.76 [s, 2H (1.99H), NH_b2_], 2.15 [s, 3H (2.98H), CH_3_].

AMB-*m*TOL: Melting point (DSC): 210/225 °C (double peak). FT-IR (KBr plate method, cm^−1^): 3442/3291 (amine N–H), 3350 (amine + amide N–H), 3026 (C_arom_–H), 2917 (C_aliph_–H), 1625 (amide C=O), 1500 (1,4-phenylene). ^1^H-NMR (400 MHz, DMSO-*d*_6_, *δ*, ppm): 9.76 [s, 2H (2.00H), NHCO], 7.70–7.62 [m, 8H (8.07H), 3,3′,5,5′-protons of AN + 3,3′,5,5′-protons of BP], 7.01 [d, 2H (1.99H), *J* = 8.3 Hz, 6,6′-protons of BP], 6.66 [d, 2H (1.98H), *J* = 8.3 Hz, 6,6′-protons of AN), 5.53 [s, 4H (3.99H), NH_2_], 2.13 [s, 6H (6.07H), 2,2′-CH_3_ of BP], 2.02 [s, 6H (5.89H), 2,2′-CH_3_ of AN]. Elemental analysis, Anal. Calcd (%) for C_30_H_30_O_2_N_4_ (478.59): C, 75.29; H, 6.32; N, 11.71. Found: C, 74.67; H, 6.46; N, 11.55.

PDA-*t*CHDCA: Another type of bisamide-diamine, PDA-*t*CHDCA (Figure 1), was also synthesized in this study. The detailed synthetic procedures and analytical data of this diamine are described in the Appendix A.

##### Monoamide-Type Diamines

Monoamide-type diamines were synthesized according to the reaction scheme shown in Figure 1. As an example of the monoamide-type diamines, the detailed synthetic procedures used for MeO-DABA are described below.

MeO-DABA: In a sealed flask, 2-methoxy-4-nitroaniline (MeO-NAN, 20 mmol) was dissolved in anhydrous THF (8.7 mL) in the presence of pyridine (2.0 mL) as an HCl acceptor. In another sealed flask, 4-NBC (22 mmol) was dissolved in anhydrous THF (10.6 mL). To the MeO-NAN solution cooled at 0 °C, the 4-NBC solution was slowly added using a syringe with continuous magnetic stirring, after which the reaction mixture was stirred at 0 °C for several hours, then subsequently at room temperature for 12 h. The formed precipitate was collected by filtration and washed with a small quantity of THF, a large quantity of water to remove the pyridine-HCl salt, and finally a small quantity of methanol. The yellowish product was dried at 120 °C for 12 h under vacuum (yield: 91%). The crude product was recrystallized from a mixed solvent (*N*,*N*-dimethylformamide (DMF)/toluene, 3:2, *v/v*). The product was confirmed to be the desired dinitro compound via the following data. Melting point (DSC): 219 °C. FT-IR (KBr plate method, cm^−1^): 3411 (amide N–H), 3105/3068 (C_arom_–H), 2993/2951 (C_aliph_–H), 1684/1550 (amide C=O), 1523/1339 (NO_2_), 1496 (1,4-phenylene). ^1^H-NMR (400 MHz, DMSO-*d*_6_, *δ*, ppm): 10.16 [s, 1H (1.01H), NHCO], 8.37 [d, 2H (1.99H), *J* = 8.8 Hz, 2,6-protons of the 4-carbonylnitrobenzene (NB) unit], 8.24–8.17 [m, 3H (3.01H), 3,5-protons of the 4-carbonyl NB unit + 5-proton of the 3-methoxy NB unit], 7.96 [dd, 1H (1.00H), *J* = 8.8, 2.5 Hz, 6-proton of the 3-methoxy NB unit], 7.89 [sd, 1H (1.02H), *J* = 2.5 Hz, 2-proton of the 3-methoxy NB unit], 4.01 [s, 3H (3.01H), OCH_3_].

The catalytic reduction of the dinitro compound was carried out as follows. The dinitro compound (15.3 mmol) was dissolved in DMF (50 mL) in the presence of Pd/C (0.487 g). The reaction mixture was refluxed at 80 °C for 5 h in a hydrogen atmosphere and cooled to room temperature without precipitation of the product. After the catalyst residue was removed by filtration, the filtrate was concentrated by an evaporator and slowly poured into a large quantity of water. The precipitate formed was washed with water and methanol and dried at 120 °C for 12 h under vacuum (yield: 73%). The pale-pink product obtained was confirmed to be the desired diamine compound (MeO-DABA) from the following data. Melting point (DSC): 144/164 °C (double peak). FT-IR (KBr plate method, cm^−1^): 3422/3249 (amine N–H), 3386/3340 (amine + amide N–H), 3034 (C_arom_–H), 2999 (C_aliph_–H), 1619/1519 (amide C=O), 1499 (1,4-phenylene). ^1^H-NMR (400 MHz, DMSO-*d*_6_, *δ*, ppm): 8.65 [s, 1H (0.99H), NHCO], 7.63 [d, 2H (2.00H), *J* = 8.6 Hz, 3,5-protons of the 4-carbonylaniline (AN) unit], 7.22 [d, 1H (1.00H), *J* = 8.4 Hz, 5-proton of the 3-methoxy AN unit], 6.56 [d, 2H (2.03H), *J* = 8.6 Hz, 2,6-protons of the 4-carbonyl AN unit], 6.28 [sd, 1H (1.00H), *J* = 2.2 Hz, 2-proton of the 3-methoxy AN unit], 6.12 [dd, 1H (1.00H), *J* = 8.4, 2.2 Hz, 6-proton of the 3-methoxy AN unit], 5.65 [s, 2H (2.01H), NH_2_ of the 4-carbonyl AN unit], 5.01 [s, 2H (2.02H), NH_2_ of the 3-methoxy AN unit], 3.70 [s, 3H (3.00H), OCH_3_]. Elemental analysis, Anal. Calcd (%) for C_14_H_15_O_2_N_3_ (257.29): C, 65.35; H, 5.88; N, 16.33. Found: C, 65.23; H, 5.93; N, 16.28.

M-DABA: Methyl-substituted monoamide-type diamine analogue was synthesized from 2-methyl-4-nitroaniline and 4-NBC in a similar manner. Melting point (DSC): 133 °C. FT-IR (KBr plate method, cm^−1^): 3424/3278 (amine N–H), 3397/3319 (amine + amide N–H), 3032 (C_arom_–H), 2925 (C_aliph_–H), 1630 (amide C=O), 1499 (1,4-phenylene). ^1^H-NMR (400 MHz, DMSO-*d*_6_, *δ*, ppm): 10.46 [s, 1H (1.01H), NHCO], 7.66 [d, 2H (1.99H), *J* = 8.6 Hz, 3,5-protons of the 4-carbonyl AN unit], 6.84 [d, 1H (1.00H), *J* = 8.3 Hz, 5-proton of the 3-methyl AN unit], 6.56 [d, 2H (1.99H), *J* = 8.6 Hz, 2,6-protons of the 4-carbonyl AN unit], 6.42 [sd, 1H (0.99H), *J* = 2.3 Hz, 2-proton of the 3-methoxy AN unit], 6.37 [dd, 1H (1.00H), *J* = 8.2, 2.5 Hz, 6-proton of the 3-methoxy AN unit], 5.63 [s, 2H (2.00H), NH_2_ of the 4-carbonyl AN unit], 4.89 [s, 2H (2.00H), NH_2_ of the 3-methoxy AN unit], 2.41 [s, 3H (3.04H), CH_3_].

##### Asymmetric Amide-Imide-Type Diamine

An asymmetric amide-imide-type diamine (PDA-HTI-DABA) was synthesized according to the reaction schemes shown in Figure 2 as follows. In a sealed flask, 4-aminobenzoic acid (20 mmol) was dissolved in anhydrous DMAc (32 mL). To this solution, 1*-exo*,2-*exo*,4-*exo*-cyclohexanetricarboxylic anhydride (H-TMA, 24 mmol) powder was gradually added, and the reaction mixture was stirred at room temperature for 6 h. A mixture of acetic anhydride (Ac_2_O, 9.4 mL) and pyridine (4 mL) was then added and stirred at room temperature for 12 h in the sealed flask. The reaction mixture was slowly poured into a large quantity of water. The white precipitate formed was collected by filtration and dried at 120 °C under vacuum (yield: 79%). Melting point (DSC): 332 °C. FT-IR (KBr plate method, cm^−1^): 2973/2931 (C_aliph_–H), 2594 (hydrogen-bonded COOH, O–H), 1779 (imide C=O), 1717 (imide C=O + hydrogen-bonded COOH, C=O), 1384 (imide N–C_arom_). ^1^H-NMR (400 MHz, DMSO-*d*_6_, *δ*, ppm): 7.98 [d, 2H (1.99H), *J* = 8.4 Hz, 2,6-protons of the terminal benzoic acid unit (BA)], 7.31 [d, 2H (2.00H), *J* = 8.4 Hz, 3,5-protons of BA], 3.27–1.30 [m, 9H (9.01H), protons of the cyclohexane unit (CHx)].

In a three-necked flask, the dicarboxylic acid obtained (3.96 g) was converted into the corresponding bis(carbonyl chloride) with thionyl chloride (30 mL) in the presence of three drops of DMF as a catalyst by refluxing at 80 °C for 4 h. After the reaction, the excess of thionyl chloride was azeotropically distilled off with benzene under a reduced pressure at 40 °C. The obtained white solid (12.5 mmol) was dissolved in anhydrous THF (11.5 mL) in a sealed flask. To this solution cooled at 0 °C, the THF solution of 4-nitroaniline (27.5 mmol) and pyridine (2.4 mL, 30 mmol) was slowly added with a syringe. The reaction mixture was stirred at 0 °C for several hours and subsequently at room temperature for 12 h. The formed yellow precipitate was washed with a small quantity of THF, a large quantity of water and ethanol, and dried at 100 °C for 12 h under vacuum (yield: 72%). FT-IR (KBr plate method, cm^−1^): 3341 (amide N–H), 3114/3086 (C_arom_–H), 2938/2865 (C_aliph_–H), 1775/1712 (imide C=O), 1683 (amide C=O), 1543/1334 (NO_2_), 1500 (1,4-phenylene). ^1^H-NMR (400 MHz, DMSO-*d*_6_, *δ*, ppm): 10.90 [s, 1H (1.00H), C_arom_–CONH], 10.58 [s, 1H (1.00H), C_aliph_–CONH], 8.29 [d, 2H (2.00H), *J* = 8.3 Hz, 2,6-protons of C_arom_–CONH–NB], 8.23 [d, 2H (2.00H), *J* = 9.3 Hz, 2,6-protons of C_aliph_–CONH–NB], 8.11–8.06 [m, 4H (4.01H), 2,3,5,6-protons of imide N–Ph], 7.85 (d, 2H (1.99H), *J* = 9.3 Hz, 2,6-protons of C_aliph_–CONH–NB], 7.54 [d, 2H (2.00H), *J* = 8.6 Hz, 3,5-protons of C_arom_–CONH–NB], 3.18–1.46 [m, 9H (8.01H), protons of CHx].

The dinitro compound obtained (9.66 mmol) was dissolved in DMF (52 mL) in the presence of Pd/C (0.538 g), and the reaction mixture was refluxed at 70 °C for 5 h in a hydrogen atmosphere. The Pd/C residue was removed by hot filtration. The filtrate was concentrated by an evaporator and slowly poured into a large quantity of water. The white precipitate formed was washed with water and methanol and dried at 120 °C for 12 h under vacuum (yield: 92%). The product was confirmed to be the desired diamine (PDA-HTI-DABA) by following data (Figure 2). Melting point (DSC): 274 °C. FT-IR (KBr plate method, cm^−1^): 3353 (amine + amide N–H), 3124/3042 (C_arom_–H), 2937/2866 (C_aliph_–H), 1774/1704 (imide C=O), 1655/1541 (amide C=O), 1516 (1,4-phenylene). ^1^H-NMR (400 MHz, DMSO-*d*_6_, *δ*, ppm): 9.96 [s, 1H (1.00H), C_arom_–CONH], 9.41 [s, 1H (1.00H), C_aliph_–CONH], 8.01 [d, 2H (2.00H), *J* = 8.5 Hz, 2,6-protons of the benzanilide unit (BzA], 7.45 [d, 2H (2.00H), *J* = 8.5 Hz, 3,5-protons of BzA], 7.38 [d, 2H (2.00H), *J* = 8.7 Hz, 2′,6′-protons of BzA], 7.21 [d, 2H (2.00H), *J* = 8.8 Hz, 3,5-protons of AN], 6.55 [d, 2H (2.00H), *J* = 8.7 Hz, 3′,5′-protons of BzA], 6.49 [d, 2H (1.98H), *J* = 8.8 Hz, 2,6-protons of AN], 4.93 [s, 4H (4.00H), NH_2_], 3.13–1.39 [m, 9H (8.00H), protons of CHx].

#### 2.1.2. Common Monomers

The structures and abbreviations of the common monomers (tetracarboxylic dianhydride and diamines) used in this study are shown in Figure 3. The sources, abbreviations, and melting points of the common monomers and raw materials used in this study are listed in Appendix A, respectively.

#### 2.1.3. Polymerization and Preparation of PI Films

Figure 4 shows three different pathways applicable to the polymerization and PI film preparation: (1) conventional two-step process (manufacturing route: T), including polyaddition, solution casting of the resultant PI precursors (poly(amic acid)s, PAAs), and thermal imidization of the PAA films; (2) chemical imidization process (manufacturing route: C), including polyaddition of PAAs, chemical imidization in the solutions, isolation of PIs, re-dissolution of the PI powder in a fresh solvent, and the formation of PI films by solution casing; (3) one-pot polycondensation process (manufacturing route: R), which is conducted by refluxing the reaction mixtures and subsequent solution casting of the PI solutions. 

In this study, polymerization was mainly conducted via the one-pot process in a four-necked separable flask, equipped with a dry nitrogen gas inlet, an outlet connected to a silicone-oil-sealed bubbler, a condenser, Dean–Stark trap, and a sealing mixer (sealed mechanical stirrer) (Nakamura Scientific Instruments Industry, UZ-SM1), with perfect sealability based on a non-contact magnetic coupling mechanism between an inner stirring rod and outer magnetic rotor. Unless otherwise stated, the polymerization was performed at an initial total monomer content of 30 wt% in this reactor via the one-pot process, with gradual dilution as appropriate with the same solvent to ensure effective mixing. In this study, the one-pot process was modified to enhance the molecular weights of the resultant PIs. The detailed procedures and reaction conditions of the modified one-pot process will be described later. 

After the one-pot polymerization, the completion of imidization was confirmed by the ^1^H-NMR spectra (DMSO-*d*_6_) from the complete disappearance of the proton signals of the PAA-inherent groups, i.e., COOH (chemical shift, *δ* = 12–13 ppm) and the NHCO groups (*δ* ~ 10.0 ppm), as shown in Figure 5. A typical FT-IR transmission spectrum of a thin film for a PI obtained via the modified one-pot process is shown in Figure 6. The spectrum included the specific bands (cm^−1^): 3368 (amide, N–H, originating from the AB-44ODA-based unit), 3068 (C_arom_–H), 2937 (C_aliph_–H), 1779/1714 (imide C=O), 1660/1536 (amide C=O from the AB-44ODA-based unit), 1500 (1,4-phenylene), 1384 (imide N–C_aliph_), 1219 (ether C_arom_–O), and 754 (imide ring deformation).

The homogeneous PI solutions obtained using the one-pot process were appropriately diluted with the same solvent for the subsequent solution casting or the isolation process, which was performed by gradually pouring the solutions into a large quantity of methanol or its aqueous solution to remove the catalyst. The isolated fibrous powder was re-dissolved in a fresh solvent for the subsequent solution casting. The typical conditions for the solution casting were as follows. The PI solutions were coated on a glass substrate and dried at 80 °C for 2 h in an air convection oven, then subsequently at 150 °C for 0.5 h + 200 °C for 0.5 h + 250 °C for 1 h under vacuum on the substrate. After being peeled off from the substrate, the PI films (typically 20 μm thick) were annealed at 250 °C for 1 h under vacuum to remove residual stress. In certain cases, the thermal conditions were fine-tuned to obtain a better quality of PI films.

In this work, the chemical compositions of the PAA and PI systems are represented with the abbreviations of the monomer components used (tetracarboxylic dianhydrides (A) and diamines (B)) as A/B for homopolymers and A/B1;B2 for copolymers.

### 2.2. Measurements

#### 2.2.1. Inherent Viscosities and Molecular Weights

The reduced viscosities (*η*_red_) of PIs or the corresponding PAAs, which can be practically regarded as their *η*_inh_, were measured in the same solvents as those used in the polymerization at a solid content of 0.5 wt% at 30 °C on an Ostwald viscometer. The number (*M*_n_) and weight (*M*_w_) average molecular weights of highly soluble PIs in THF were determined by gel permeation chromatography (GPC) using THF as an eluent at room temperature on a Jasco LC-2000 Plus HPLC system with a GPC column (Shodex, KF-806L) at a flow rate of 1 mL min^−1^ by ultraviolet-visible detection at 300 nm (Jasco, UV-2075). The calibration was performed using standard polystyrenes (Shodex, SM-105).

#### 2.2.2. Linear Coefficients of Thermal Expansion (CTE)

The CTE values of PI specimens (15 mm long, 5 mm wide, and typically 20 μm thick) in the *X*–*Y* direction below the *T*_g_s were measured by thermomechanical analysis (TMA) as an average in the range of 100–200 °C at a heating rate of 5 °C min^−1^ on a thermomechanical analyzer (Netzsch, TMA 4000) with a fixed load (0.5 g per unit film thickness in μm, i.e., 10 g load for 20-μm-thick films) in a dry nitrogen atmosphere. In this case, after the preliminary first heating run up to 120 °C and successive cooling to room temperature in the TMA chamber, the data were collected from the second heating run to remove the influence of adsorbed water. 

#### 2.2.3. Heat Resistance

The glass transition temperatures (*T*_g_s) of PI films were determined from the peak temperatures of the loss of energy (*E*″) curves by dynamic mechanical analysis (DMA) at a heating rate of 5 °C min^−1^ on the TMA instrument (as before). The measurements were conducted at a sinusoidal load frequency of 0.1 Hz with an amplitude of 15 gf in a nitrogen atmosphere.

The thermal and thermo-oxidative stability of PI films were evaluated from the 5% weight loss temperatures (*T*_d_^5^) by thermogravimetric analysis (TGA) on a thermo-balance (Netzsch, TG-DTA2000). TGA was performed by heating the sample-charged open aluminum pan from 30 to 500 °C at a heating rate of 10 °C min^−1^ in a dry nitrogen or air atmosphere. A small amount of weight loss due to the desorbed water around 100 °C during the TGA heating runs was compensated for by off-setting at 150 °C to 0% for the data analysis.

#### 2.2.4. Optical Transparency

The light transmission spectra of the PI films (typically 20 μm thick) were measured on an ultraviolet–visible spectrophotometer (Jasco, V-530) in the wavelength (*λ*) range of 200–800 nm. The light transmittance at 400 nm (*T*_400_) and the cut-off wavelength (*λ*_cut_) at which the transmittance becomes substantially zero were determined from the spectra. The yellowness indices (YI, ASTM E 313) for PI films were determined from the spectra under a standard illuminant of D65 and a standard observer function of 2° using color calculation software (Jasco) on the basis of the following relationship:YI = 100 (1.2985*x* − 1.1335*z*)/*y*
(1)
where *x*, *y*, and *z* are the CIE tristimulus values. YI takes zero for an ideal white or transparent sample. The total light transmittance (*T*_tot_, JIS K 7361-1) and the diffuse transmittance (*T*_diff_, JIS K 7136) of PI films were measured on a double-beam haze meter equipped with an integrating sphere (Nippon Denshoku Industries, NDH 4000). The haze levels (turbidity) of PI films were calculated from the relationship:Haze = (*T*_diff_/*T*_tot_) × 100(2)

#### 2.2.5. Birefringence

The in-plane (*n*_in_ or *n*_xy_) and out-of-plane (*n*_out_ or *n*_z_) refractive indices of PI films were measured with a sodium lamp at 589.3 nm (*D*-line) on an Abbe refractometer (Atago, 4T, *n*_D_ range: 1.47–1.87) equipped with a polarizer using a contact liquid (sulfur-saturated methylene iodide *n*_D_ = 1.78–1.80) and a test piece (*n*_D_ = 1.92). The birefringence of PI films, which represents the relative extent of chain alignment in the *X*–*Y* direction, was calculated from the following relationship: Δ*n*_th_ = *n*_in_ − *n*_out_(3)

#### 2.2.6. Mechanical Properties

The tensile modulus (*E*), tensile strength (*σ*_b_), and elongation at break (*ε*_b_) of PI specimens (film dimension: 30 mm long, 3 mm wide, typically 20 μm thick; specimen numbers > 15) were measured on a mechanical testing machine (A&D, Tensilon UTM-II) at a cross head speed of 8 mm min^−1^ at room temperature. The specimens were cut from high-quality film samples (10 cm × 10 cm) free of any defects such as fine bubbles. The data analysis was carried out on a data processing program (Softbrain, UtpsAcS Ver. 4.09). 

#### 2.2.7. Water Uptake

The degrees of water absorption (*W*_A_, %) of the PI films were determined according to the JIS K 7209 standard using Equation (4): *W*_A_ = [(*W* − *W*_0_)/*W*_0_] × 100(4)
where *W*_0_ is the weight of a film sample just after vacuum-drying at 50 °C for 24 h and *W* is the weight of a film immersed in water at 23 °C for 24 h and carefully blotted dry with tissue paper. In this case, large film specimens (>0.1 g) were used to minimize the experimental error.

#### 2.2.8. Surface Hardness

The surface hardness of the PI films was estimated by pencil scratch test using a pencil hardness tester (BEVS, 1301/750). This apparatus consists of a cylindrical pencil lead with a load of 750 g, which was polished using sandpaper to obtain a smooth flat surface, according to the ASTM D3363. In the test, standard pencils of differing hardness (Mitsubishi Pencil Uni (6B–6H) were selected and mounted on the tester at a fixed angle of 45°. After running the tester for at least 7 mm at a moving speed of 0.5–1.0 mm s^−1^, the presence or absence of a scratch or indentation on the film surface was observed under an optical microscope. For this test, the film specimens were generally attached to a substrate using double-sided adhesive tape. The problem with this procedure, however, was that the smooth slippage of the pencil tip was frequently disturbed by local “slackening” in the attached films during the scratch test. This meant that the attached specimens were easily scratched, even with a low-hardness pencil. Therefore, the test was performed using coatings directly formed on glass substrates to avoid this underestimation of the surface hardness.

## 3. Results and Discussion

### 3.1. Importance of Polymerization, Imidization, and Film Preparation Routes

Our initial aim was to assess whether the goal of this study is achievable by a simple combination of H-PMDA and diamines with rigid and linear structures. 2,2′-Bis(trifluoromethyl)benzidine (TFMB) is empirically the optimal diamine for simultaneously achieving low CTE, high transparency, and excellent solubility in the resultant PIs. However, our first attempt to obtain a free-standing flexible H-PMDA/TFMB PI film was unsuccessful because the film obtained via the conventional two-step process included many cracks. The generation of these cracks was probably the result of poor chain entanglement due to insufficient molecular weight of the PAA (*η*_red_ = 0.25 dL g^−1^, Figure 7). Similarly, solution casting via chemical imidization (process C, Figure 4) afforded a highly cracked film because the molecular weights did not change significantly during chemical imidization [52]. In addition, even when a commonly used highly reactive aromatic diamine, viz. 4,4′-ODA, was used instead of TFMB, the *η*_red_ value of the resultant PAA was not dramatically enhanced (0.60 dL g^−1^), as shown in Figure 7. 

On the other hand, CBDA easily afforded a PAA with a considerably high *η*_red_ value (>1.5 dL g^−1^) by reacting with TFMB. This extremely high reactivity was probably closely related to the strain accumulated in the acid anhydride ring connected to the central cyclobutane unit in CBDA [18,23]. In contrast, another commercially available cycloaliphatic tetracarboxylic dianhydride, viz. bicyclo(2.2.2)oct-7-ene-2,3,5,6-tetracarboxylic dianhydride (BTA), provided PAAs with lower *η*_red_ values than those of H-PMDA, as shown in Figure 7. In general, BTA-based PIs are thermally less stable than H-PMDA-based counterparts because BTA undergoes the retro Diels–Alder reaction at elevated temperatures [53].

Thus, H-PMDA has a better chance of obtaining practically useful colorless PI films with higher ductility and thermal stability than BTA if an effective way to dramatically enhance the molecular weights of the PAAs or PIs is available. These results clarified our first issue to be addressed, i.e., to establish an effective polymerization process for significantly increasing the molecular weights and ensuring sufficient film-forming ability in the H-PMDA/TFMB and other H-PMDA-based systems.

### 3.2. Strategy 1: Modification of the One-Pot Process

A possible approach to ensuring sufficient film-forming ability of the H-PMDA/TFMB system is to modify the conventional one-pot process. An expected benefit of this approach is that the film formation process, involving direct casting of the PI solutions obtained after the one-pot process, is simple and advantageous for reducing the CTEs of the resultant PI films. Here, the generated low CTE is attributed to the spontaneous in-plane chain orientation induced during solution casting [30].

#### 3.2.1. Reaction Temperature Profile

The reaction temperature profile of the conventional one-pot process, which comprises two separate reaction processes, viz. polyaddition and subsequent imidization in solution, is schematically shown in Figure 8a. In general, the polyaddition is first conducted without heating to maintain a high equilibrium constant (*K*_PAA_) (because it is an exothermic reaction), after which the reaction mixtures are refluxed at 160–220 °C for several hours to complete imidization. In this second reaction process, partial depolymerization of the PAAs can occur with imidization while increasing the reaction temperature [54,55], and the impact of depolymerization on the final molecular weight becomes prominent when the initial molecular weight of PAA is high.

Contrary to this general procedure, our modified one-pot process involved rapidly increasing the reaction temperature to 200 °C immediately after complete addition of H-PMDA powder to the diamine solutions (rapid heating process), followed by holding at 200 °C for 4 h in a nitrogen atmosphere, as shown in Figure 8b. In this case, the polyaddition and imidization proceed simultaneously. This suggests that the concentration of amic acid (AA) units in the reaction mixture remains quite low, even in the early reaction stage. Therefore, this situation can be thermodynamically advantageous to suppress the retro acylation of the remaining AA units. If this effect overcomes the predicted negative effect of rapid heating (a decrease in the *K*_PAA_ at elevated temperatures), the proposed rapid heating process can ultimately be effective in enhancing the molecular weights of the resultant PIs.

This rapid heating process (Figure 8b) is advantageous in terms of saving time compared with the conventional process (Figure 8a).

#### 3.2.2. Solvents

Phenolic solvents (e.g., *p*-chlorophenol and *m*-cresol) are often used in laboratory-scale one-pot polycondensation processes because of their benefits (high dissolution ability for various PIs and suitability for enhancing the molecular weights of the resultant PIs) [56,57]. In addition, it has been previously reported that the use of *m*-cresol led to a more flexible copolyimide film than that obtained via a two-step process using common amide solvents [57]. However, these phenolic solvents are unsuitable for industrial applications because of their toxicity and low volatility (b.p. 220 °C for *p*-chlorophenol, 203 °C for *m*-cresol), which make them undesirable for direct solution casting. 

However, the use of amide solvents (particularly NMP) often causes appreciable coloration of the solutions during the one-pot process, resulting in the coloration of PI films. In contrast, *γ*-butyrolactone (GBL) does not significantly contribute to coloration, unlike amide solvents, although GBL generally has a slightly lower dissolution power for various monomers and polymers generated than amide solvents. Therefore, in this study, GBL was mainly selected as the optimal solvent for the one-pot polymerization process. In addition, GBL has a boiling point (204 °C) high enough for complete imidization and sufficient volatility for the subsequent solution casting.

#### 3.2.3. Azeotropic Reagents

Azeotropic reagents are generally added to the reaction mixtures to remove the water generated by imidization during the one-pot process. However, in some cases, the addition of azeotropic reagents (typically benzene or toluene) contributed to precipitation or gelation during the one-pot process because these also behave as poor solvents for the PIs yielded. This is a significant obstacle, particularly for PI systems that do not have extremely high solubility. 

However, no adverse influence on the reaction was observed in the H-PMDA/TFMB system, even in the absence of toluene. Therefore, in this study, in contrast to the general procedure, one-pot polymerization was conducted without using azeotropic reagents. The original dissolution ability of GBL, maintained by the absence of toluene, greatly hindered the precipitation during the one-pot process.

#### 3.2.4. Catalysts

Less volatile basic organic compounds (typically quinolone, isoquinoline, and *γ*-picoline) are generally added to the reaction mixture as imidization-promoting catalysts. The low volatility of these catalysts is desirable to avoid the disappearance of a major portion of the added catalysts in the initial stage of the one-pot reaction at elevated temperatures. However, less volatile catalysts easily remain in the films during subsequent solution casting. Hence, to avoid this issue, 1-ethylpiperidine (1-EP, b.p. 131 °C) with higher volatility was selected as the catalyst in this study. To suppress the disappearance of 1-EP during the one-pot process, the aforementioned reactor with extremely high sealability was used. 

As mentioned later, the use of 1-EP was effective in enhancing the molecular weights of the resultant PIs. In addition, an increase in the 1-EP content resulted in a monotonous increase in molecular weight, which was accompanied by an increase in the coloration of the reaction mixture. Therefore, in this study, 1-EP was added at a controlled molar ratio ([1-EP]/[theoretical dehydration due to imidization] = 1/1) to obtain high-molecular-weight PIs while suppressing coloration.

#### 3.2.5. Effects of Modified One-Pot Polymerization Process

Figure 9 shows the effect of the modified one-pot process for the H-PMDA/TFMB system. Whereas the PAA obtained by polyaddition at room temperature and its chemically imidized sample did not exhibit any film-forming ability because of their considerably low molecular weights, the aforementioned rapid heating of the monomer mixture to 200 °C in GBL at an initial monomer content of 50 wt% without catalyst and subsequent holding at 200 °C for 4 h produced a homogeneous and viscous solution of the PI with an appreciably increased *η*_red_ value (0.74 dL g^−1^) and sufficient film-forming ability. In addition, the use of 1-EP as an imidization-promoting catalyst was more effective in enhancing the molecular weight (*η*_red_ = 1.24 dL g^−1^, *M*_n_ = 5.80 × 10^4^, and *M*_w_ = 1.97 × 10^5^). The prominent effect of 1-EP can be attributed to the low-temperature shift in the temperature range of imidization, thereby enhancing the *K*_PAA_ compared with that in the catalyst-free system.

The modified one-pot reaction was also applied to a system comprising H-PMDA and another rigid diamine, viz. *m*-tolidine (*m*-TOL). This system produced a homogeneous and viscous solution of the PI with a sufficiently high molecular weight, and the cast film was sufficiently ductile. The good solubility of this PI in GBL probably results from the combined effect of a highly twisted (non-coplanar) conformation in the biphenylene units, arising from the steric hindrance between the 2,2′-disubstituted methyl groups and 6,6′-hydrogen atoms in the biphenylene unit [58], as well as the non-coplanar and bent steric structure of H-PMDA-based diimide units [41].

### 3.3. Properties of H-PMDA-Based PIs Obtained from Common Diamines 

#### 3.3.1. Effects of Rigid Diamines with Substituents

Table 1 summarizes the film properties of the PIs obtained from H-PMDA and various common diamines via the modified one-pot process. The use of ether-containing common diamines (samples #1–3) resulted in highly transparent and ductile PI films. The H-PMDA/4,4′-ODA film (#1) was particularly tough, as suggested by its extremely high *ε*_b_ value. However, these ether-containing PI films did not exhibit low CTEs, as predicted from their highly bent (non-linear) backbone structures, which are disadvantageous for inducing significant main-chain alignment in the *X*–*Y* direction (in-plane orientation) during solution casting. The pencil hardness of the film (#1) directly formed on the glass substrate was estimated to be 3H.

On the other hand, the H-PMDA/TFMB system (#4) without flexible ether linkages was investigated with the expectation of low CTE generation. However, this PI film did not exhibit low CTE (57 ppm K^−1^), even though other desired properties (excellent optical transparency, a considerably high *T*_g_, and the requisite degree of ductility) were maintained. The pencil hardness of the film (#4) directly formed on a glass substrate was estimated to be 4H, although it was significantly underestimated to be 5B when the films attached to a substrate with a double-sided adhesive tape were used. The *W*_A_ of the PI film was 1.14%. Similarly, the use of another typical rigid diamine, *meta*-tolidine (*m*-TOL, #5), was ineffective in reducing the CTE, although it also had other excellent combined properties, as shown in Table 1. 

None of the PI films listed in Table 1 were broken in a 180° bending test at a zero-curvature radius. 

Thus, it was difficult to achieve low-CTE characteristics when a twisted structure of rigid diamines with bulky substituents, such as TFMB and *m*-TOL, was used. 

#### 3.3.2. Use of Rigid Diamines without Substituents

In the next approach, a typical rigid diamine without substituents, *p*-phenylenediamine (*p*-PDA, #7), was used with the expectation of low CTE generation. However, this attempt failed because precipitation occurred during the modified one-pot process (Table 2). Similarly, another non-substituted rigid diamine, ester-linked APAB (#15), was not compatible with this reaction process because of precipitation.

DABA is the preferred diamine for the present purpose because of its structural rigidity owing to its double-bond character based on keto-enol tautomerization, which is favorable for low CTE generation, as well as the expected affinity of the amide group with the solvent, which is favorable for modified one-pot process compatibility. Contrary to the expectations, the reaction between DABA and H-PMDA (#11) caused precipitation. The great difficulty in this challenge is attributed to the trade-off between low CTE and excellent solubility in GBL (process compatibility).

In all of these systems, the reaction mixtures became temporally homogeneous in the early stages of the reaction. Therefore, the precipitation or gelation generated in the later stages of the reaction was probably due to the increased molecular weights of the PIs and practically complete imidization during the one-pot process. 

The aforementioned individual results for the TFMB- and *p*-PDA-derived systems suggest that copolymerization using these diamines can be applied. With decreases in *p*-PDA content from 100, 50, 30, and 20 mol% in the copolymers (#7–10), the appearance of the reaction mixture after the one-pot process changed from precipitation to gelation, accompanied by a decrease in the turbidity, corresponding to a gradual improvement in solubility of the resultant PIs. Similar behavior was observed for the H-PMDA/DABA;4,4′-ODA copolymers; with decreases in DABA content from 100, 70, and 60 mol% in the copolymers (#11–13), the amount of precipitate was gradually reduced (Table 2). 

On the other hand, the copolyimide system (#6) comprising H-PMDA with DABA (70 mol%) and TFMB (30 mol%) temporarily afforded a homogeneous solution immediately after the reaction; gelation occurred after standing at room temperature for one day. However, gelation disappeared owing to warming. Casting from its homogeneous solution afforded an almost colorless and ductile film with a considerably high *T*_g_ and relatively low CTE (41 ppm K^−1^), as listed in Table 1. 

### 3.4. Strategy 2: Modification of the Diamine Structures

The abovementioned results suggest that as long as the existing monomers such as DABA, TFMB, and *m*-TOL were used, there was a limit to simultaneously achieving low CTEs and other target properties using the modified one-pot method. To date, we have studied a series of amide-containing PIs and their unique properties, such as the self-in-plane orientation behavior during solution casting [47,51,52]. Based on the knowledge obtained, as the next strategy, we structurally modified the diamines for combination with strategy 1 (modified one-pot method). 

#### 3.4.1. Effects of Monoamide-Type Diamines Modified by Substituents

Our initial approach was to modify DABA with substituents. The reaction of H-PMDA and methyl-substituted DABA (M-DABA, #14) temporarily led to a homogeneous solution; however, gelation occurred soon during standing at room temperature after the reaction was completed (Table 2). Thus, this system was not suitable for the one-pot process in terms of the insufficient solution stability. However, this result was not consistent with that mentioned in a previous report [28]; there was no description of gelation after a similar one-pot polymerization of M-DABA and hydrogenated PMDA (its steric structure is unknown), and the resultant solution provided a colorless PI film with a relatively low CTE. This contradiction can be attributed to the differences in the steric structures of the hydrogenated PMDA and our H-PMDA (1-*exo*,2-*exo*,4-*exo*,5-*exo*-cyclohexanetetracarboxylic dianhydride), the molecular weights, and the extent of imidization in the resultant PI. 

In contrast, the use of MeO-DABA (#19), which contains a methoxy group, was effective in preventing precipitation during the reaction, and a highly stable PI solution was obtained. This system afforded a highly ductile film with relatively high optical transparency and relatively low CTE (40.8 ppm K^−1^), as listed in Table 3. The slightly reduced optical transparency was probably due to trace amounts of residual unknown colored impurities in MeO-DABA. Thus, the effect of the methoxy substituent was unexpectedly large in terms of improving the solubility in GBL, although there was still room for further improvement in the low-CTE characteristics.

#### 3.4.2. Effects of Amide-Imide-Type Diamine

Our next attempt was to obtain compatibility with the modified one-pot process and low CTEs for the resultant PIs using the designed diamines without substituents. PIs with asymmetric local structures have been investigated [59,60], and they frequently exhibit excellent solubility owing to their highly distorted structures. In this study, we designed another type of asymmetric diamine (PDA-HTI-DABA, #20) in which a structural linearity or rigidity is maintained. Despite the absence of substituents such as methoxy groups, this asymmetric diamine led to a homogeneous and highly stable PI solution after the reaction. The cast film exhibited a relatively low CTE (43.9 ppm K^−1^) while maintaining an extremely high *T*_g_, excellent optical transparency, and sufficient ductility with resistance to the 180° bending test, as shown in Table 3.

#### 3.4.3. Effects of Bisamide-Type Diamines

Finally, the effects of bisamide-type diamines were investigated. The properties of the PI-cast films are presented in Table 4. The system obtained from AB-44ODA, #21), as well as that from AB-34ODA (#22), exhibited a relatively low CTE (37.9 ppm K^−1^), while maintaining a considerably high *T*_g_, excellent optical transparency, and extremely high *ε*_b_ value. The significant toughening effect observed in the H-PMDA/AB-44ODA system (#21) was similar to that observed in the H-PMDA/4,4′-ODA system (#1). This probably reflects the chain entanglement effect promoted by the 4,4′-diphenyl ether units. 

The use of isomeric AB-34ODA (#22) was expected to reduce the CTE because the 3,4′-linked diphenyl ether local unit can adopt a conformation with a relatively linear form compared with that of its 4,4′-linked counterpart. However, this approach was less effective in reducing the CTE than expected. This PI film was extremely tough, similar to that of the isomeric AB-44ODA-derived system (#21).

To decrease the CTE further, AB-APAB (#16), which contains a more rigid *para*-linked phenyl benzoate unit at the central portion, was used. However, there was no compatibility with the modified one-pot process due to precipitation (Table 2). Similarly, the use of an AB-APAB analog, methyl-substituted AB-ATAB (#17), was unsuccessful. Another type of bisamide-diamine, PDA-*t*CHDCA (#18 and #18′), was also used. However, despite the non-planar structure (chair form) of the central *trans*-1,4-cyclohexylene unit, the expected solubility-improving effect was not observed (Table 2).

On the other hand, when using AB-*m*TOL (#23), which contains a twisted *p*-biphenylene unit in the central portion, a homogeneous PI solution was readily obtained after the reaction, and this PI afforded a colorless film with a significantly reduced CTE (32.1 ppm K^−1^) and other desired properties, as listed in Table 4. The pencil hardness of the film (#23) formed directly on the glass substrate was estimated to be 4H.

An analog of AB-*m*TOL, AB-TFMB (#24 and #24′), was the most appropriate diamine to reduce the CTE (30.1 ppm K^−1^ (#24) and 25.9 ppm K^−1^(#24′)) and simultaneously enhance the optical transparency, *T*_g_, and toughness; however, this PI film had a relatively high *W*_A_ value (2.97%), which was probably inevitable owing to the high polarity of the amide group. Furthermore, the use of another analog of AB-*m*TOL, AMB-*m*TOL (#25), with an increased number of methyl substituents, achieved the lowest CTE (24.3 ppm K^−1^) in this study. This result was probably related to the reduced rotational flexibility around the imide *N*–Ar bond due to the presence of the terminal (*ortho*-positioned) methyl groups and the significantly increased molecular weight (*η*_red_ = 3.82 dL g^−1^) of this PI. This diamine contributed to further improvement of the optical transparency of the PI film compared with AB-*m*TOL. All PI films listed in Table 4 exhibited resistance to the 180° bending test.

To clarify the effects of the incorporated *p*-benzamide units (Ar–NH–CO–Ar) on the properties, the H-PMDA-based PIs obtained from bisamide-type diamines (AB-X, X: the central diamine structures) were compared with the counterparts from the corresponding common diamines. When using AB-Xs instead of the common diamines, the thermal stability (*T*_d_^5^ in N_2_) decreased slightly because of the presence of amide linkages, which are thermally less stable than the ether connecting groups (Ar–O–Ar) [61], as shown in Figure 10a. In contrast, the *T*_g_s tended to increase (Figure 10b), probably reflecting the enhanced main-chain rigidity and interchain hydrogen-bonding ability. The effect of the incorporation of *p*-benzamide units on the decrease in the CTEs was more prominent, particularly in the systems comprising twisted biphenylene structures, as shown in Figure 10c. The results were undoubtedly due to the enhanced structural linearity and stiffness. 

However, the effect on the film toughness was very complicated; the AB-*m*TOL-derived system (#23) virtually inherited the original film toughness (low *ε*_b_) of the *m*-TOL-derived counterpart (#5), similar to the relationship between the AB-44ODA-derived system (#21) and the 4,4′-ODA-derived counterpart (#1). In contrast, there was no close relationship between the toughness of the AB-34ODA-derived system (#22) and the 3,4′-ODA-derived counterpart (#3), similar to the relationship between the AB-TFMB-derived system (#24) and the TFMB-derived counterpart (#4). Thus, it was difficult to rationalize the prominent toughening behavior of these AB-X-derived systems.

### 3.5. Mechanism of Low CTE Generation during Solution Casting

#### 3.5.1. Primary Factor for Generation of Low CTEs

To ascertain whether the low CTEs observed in these H-PMDA-based systems were essential results, which were caused by a high extent of in-plane chain orientation, the correlation of the CTEs and the *Z*-direction birefringence (Δ*n*_th_), which qualitatively represents the degree of in-plane orientation, was examined. This was because irreversible film shrinkage, which gives the misleading result of an ‘apparent’ low CTE, frequently occurs during the heating run by TMA due to various factors (e.g., the relaxation of residual stress, desorption of the adsorbed water, and others, particularly the former) [30]. If the observed low CTEs primarily resulted from these non-essential factors, a good relationship between the CTE and Δ*n*_th_ would ordinarily collapse. However, in fact, a good correlation was observed, where the CTE decreased almost linearly with increasing Δ*n*_th_ (Figure 11), although strictly speaking Δ*n*_th_ also depends on the polymer structure. This suggests that the low CTEs observed in this study can undoubtedly be attributed to the high extent of in-plane chain orientation.

#### 3.5.2. Factors Influencing the In-Plane Orientation Behavior during Solution Casting 

As previously mentioned, simple solution casting does not generally serve as the driving force to promote the in-plane orientation of the polymer chains. Nonetheless, a prominent spontaneous in-plane orientation, which resulted in increased Δ*n*_th_, was observed after solution casting in certain amide-containing PIs examined in this study (Figure 11). We have reported that certain highly soluble PIs or poly(ester imide)s with rigid and linear backbone structures exhibit similar behavior and proposed that the following factors govern the generation of in-plane orientation during solution casting [24,30,47,48,49,50,51,52]: (1) the structural linearity and stiffness of the PI main chains; (2) the molecular weight of the PIs; (3) the processing conditions, involving slower drying of coatings at lower temperatures or at lower heating rates or using solvents with lower boiling points. 

The most influential factor was undoubtedly the main-chain linearity and stiffness. To discuss factor (1), the extended chain forms, which are useful for qualitatively estimating chain linearity [24], are depicted for certain selected PIs in Figure 12. There is an accepted criterion that unless the drawing of the extended chain is highly linear, the real system usually does not generate significant in-plane orientation, as a result low CTEs. In the H-PMDA/4,4′-ODA system that did not show a low CTE, the extended chain form depicted in Figure 12a appears as significantly meandering compared to that of H-PMDA/AB-44ODA (Figure 12b), which exhibited a lower CTE. Similarly, the extended chain forms of the H-PMDA/*m*-TOL and H-PMDA/TFMB systems without low CTEs (Figure 12c) appear as significantly meandering compared with those of the low-CTE H-PMDA/AB-*m*TOL and H-PMDA/AB-TFMB systems (Figure 12d), although the chain linearity of the latter is still lower than that of the CBDA-based counterparts (Figure 12e). Thus, a close relationship between the main-chain linearity, which can be estimated from the meandering nature of the extended chains, and the CTEs of the real PI films was reconfirmed in the H-PMDA-based PIs examined in this study.

The solution casting process, which is accompanied by solvent evaporation and a concomitant thickness decrease in coatings on the substrates, is schematically shown in Figure 13a. The aforementioned main-chain linearity (factor (1)) is probably related to the magnitude of the ‘apparent’ stress in the *X*–*Y* direction, which can start to be generated when the wet coatings reach a tack-free half-dried state, i.e., a glassy state with a *T*_g_ identical to the initial mild drying temperature (*T*_dry_ = 80 °C for the GBL solutions or 50 °C for the CPN solutions) (Figure 13b), which is continuously applied until drying is completed. The apparent stress, which has an apparent stretching effect, can be assumed to increase concomitantly with decreasing coating thickness from the half-dried to the completely dried state, while preventing film contraction in the *X*–*Y* direction by the coating–substrate adhesion. The expected apparent stretching effect can serve as a trigger for inducing in-plane orientation.

In addition, if liquid–crystal-like domains can be formed during solution casting, they can contribute to inducing the in-plane orientation with the help of the apparent stretching effect. This hypothesis is inspired by our previous results [62]; a partially imidized PAA of 3,3′,4,4′-biphenyltetracarboxylic dianhydride (*s*-BPDA)/*p*-PDA formed a lyotropic liquid crystal (LC), which could be observed under a polarizing microscope (POM), in NMP at a high solute content, and thermal imidization of the LC-remaining precursor cast film resulted in a significantly reduced CTE compared with the counterpart from the LC-free common PAA film (Appendix A). 

However, in the present H-PMDA-based systems with low CTEs, no optical anisotropic textures were observed after drying at 80 °C using POM. Therefore, if the locally ordered domains in our systems are much smaller than the visible wavelength range (a detection limit of POM), the proposed hypothesis is reasonable.

The effect of the molecular weight (factor (2)) was observed in samples #1, #4, #5, and #19. The increase in molecular weight (*η*_red_) contributed to a slight reduction in the CTEs, as listed in Table 1 and Table 3. This effect can be explained as follows [24]. A higher-molecular-weight PI has a higher solution viscosity, due to which the coatings can become a tack-free half-dried (glassy) state at a lower degree of dryness (with more solvent), compared with the coatings of lower-molecular-weight PIs. This causes a larger decrease in coating thickness from the half-dried state to the completely dried state, during which the coatings continue to undergo an apparent stretching effect.

The effect of the volatility of the casting solvents (factor (3)) was observed in samples #24 and #24′. The use of solvents with lower boiling points for solution casting has the same effect as the aforementioned molecular weight effect. In fact, the use of cyclopentanone (CPN, b.p. 131 °C) resulted in a slight decrease in the CTE compared with GBL (b.p. 204 °C), as listed in Table 4, although the former often causes slight coloration of the solution when heated at an elevated temperature during the dissolution of PI powder.

Here, we organized the key factors governing the generation of a low CTE and their interrelations in Figure 14. They were classified into chemical (chain linearity and stiffness and molecular weights) and physical factors (spontaneous in-plane orientation behavior). The physical factors included the aforementioned apparent stretching effect accompanied by the decreased thickness of the coatings, the expected locally ordered structure formation, the coating modulus (*E* (*C*_s_)) varying with the solvent content (*C*_s_), and the conditions of solution casting. The apparent stretching effect is likely related to the molecular weight (*M*_w_), coating *T*_g_ (*C*_s_), and coating viscosity (*η* (*C*_s_)), which increases with increasing *M*_w_ and hydrogen-bonding ability. Thus, these physical factors are probably interrelated to the chemical factors (chain structures), as shown in the yellow portions in Figure 14. There could be a similar interrelation between *E* (*C*_s_) and the main-chain linearity (i.e., *E* (*C*_s_) strongly depends on the main-chain linearity), as well as between the ease of the locally ordered structure formation and the chain linearity.

### 3.6. Performance Balance

The performance balance of the H-PMDA-based low-CTE PIs obtained in this study was reviewed using spider charts, which were evenly expanded when the target properties were comprehensively achieved. The criteria for the spider charts were established by ranking the achievement levels of the individual target properties, as shown in Table 5. Here, the realistically achievable maximum level was ranked 5, the virtually minimum level was ranked 1, and the intermediate levels were ranked 2–4 by equally dividing the span between the maximum and minimum values. For example, the *T*_g_s ≥ 360 °C were assigned to rank 5 based on the *T*_g_ (360–370 °C) of the *s*-BPDA/*p*-PDA PI film [63], which is recognized as a typical PI with the highest heat resistance. On the other hand, the *T*_g_s ≤ 200 °C were assigned to rank 1 based on the *T*_g_ of bisphenol A-type poly(ether imide), which is known as a typical thermo-processable PI with the lowest level of *T*_g_ among PIs [57,64].

Regarding the ranking of low-CTE properties, PIs with extremely low CTEs (≤10 ppm K^−1^) were assigned to rank 5 based on the CTE (5–15 ppm K^−1^ [45,46]) of *s*-BPDA/*p*-PDA, which is a typical low-CTE PI, and PIs with very high CTEs (≥70 ppm K^−1^), which are often observed in common flexible polymers, were ranked 1. 

Figure 15 shows the spider chart. The H-PMDA/TFMB system (manufacturing route: R) did not exhibit low-CTE characteristics and good toughness. Therefore, this spider chart is not evenly expanded, as shown in Figure 15a. In contrast, the CBDA/TFMB system (manufacturing route: T) exhibited a low CTE; however, it did not have sufficient solution-processability and toughness. Therefore, this chart is biased to the right (Figure 15b). On the other hand, as suggested from the evenly expanded spider chart, the H-PMDA-AB-TFMB system achieved virtually all of the target properties (Figure 15c). The H-PMDA/AMB-*m*TOL system, which was slightly superior to the H-PMDA/AB-TFMB system in terms of the low CTE property, also showed a relatively expanded spider chart (Figure 15d), although there was room for further improvement in the film toughness. The results indicate that these amide-containing H-PMDA-based PIs are promising candidates as novel heat-resistant plastic substrates for use in image display devices.

## 4. Conclusions

In this study, we aimed to develop practically useful colorless PIs with low CTEs and other desired properties by overcoming the great difficulties involved in reducing CTEs, which are ascribed to the steric structure of H-PMDA. The conventional two-step process using H-PMDA and TFMB with a rod-like structure afforded a PI without film-forming ability owing to its insufficient molecular weight, similar to the film preparation process involving chemical imidization. To significantly enhance the molecular weight, the conditions of one-pot polymerization were thoroughly modified in terms of the timing of the heating–refluxing process, heating rates, azeotropic reagents, catalyst, and reactor structure. The modified one-pot process led to a dramatically enhanced molecular weight of the H-PMDA/TFMB PI with sufficiently high film-forming ability. However, the PI film cast from its homogeneous solution did not exhibit low CTE characteristics, similar to the analogous system using *m*-TOL. 

To solve this problem, a series of amide- and amide-imide-containing diamines were synthesized. The modified one-pot polymerization of H-PMDA and these diamines led to homogeneous, viscous, and stable PI solutions with high solid contents. The resulting cast films of certain H-PMDA-based systems simultaneously achieved low CTEs, high optical transparency, considerably high *T*_g_s, and sufficient ductility. The excellent performance balance was ascertained using evenly expanded spider charts.

Herein, we propose a possible mechanism for spontaneous in-plane orientation behavior during solution casting. The generation of low CTEs is closely related to both chemical (main-chain linearity and stiffness and molecular weights) and physical factors (the degree of spontaneous in-plane orientation during solution casting).

The combined strategies of modifying the diamine structures and the polymerization process afforded certain H-PMDA-based PIs that mostly met the target properties. They are promising candidates as heat-resistant colorless plastic substrates for use in image display devices and other optical applications. 

## Data Availability

The data supporting this study are available within the article and its Appendix A.

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
