# Peer review of "Solution-Processable Colorless Polyimides Derived from Hydrogenated Pyromellitic Dianhydride: Strategies to Reduce the Coefficients of Thermal Expansion by Maximizing the Spontaneous Chain Orientation Behavior during Solution Casting"

_polymers, 2022, doi:10.3390/polym14061131_

Round 1

Reviewer 1 Report

  1. Please supplement the data you did not provide in the Table 1, Table 3 and Table 4.
  2. It would be better to present data in the form of figure.

Author Response

Q1. Please supplement the data you did not provide in the Table 1, Table 3 and Table 4.

A1. We filled in the blanks in these tables as much as possible (green-marked) except for others that we could not fill because of the absence of the film specimens left for re-measurements.

Q2. It would be better to present data in the form of figure.

A2. According to your advice, we converted a portion of the current table data to a figure form, consequently added a new figure (Figure 10) in the revised manuscript.

Reviewer 2 Report

Dear Authors,

My comments bellow:

Lack of the line numbers.
What affects low coefficient of thermal expansion?
TGA measurement: could you describe this method better? Range of temperatures etc?
Optical transparency: what means the symbol in brackets?
The goal of the work should be presented in the Introduction section instead in the Results and Conclusions part.
3.4.1. The last sentence in this paragraph is not finished.
References: should be refreshed - most of them are too old. Also mostly Japanese authors are cited. It should be a balance with other authors who publish in this field as well, unless Japanese research dominates in this subject.
Did you take photographs of these PIs?
Lack of explanation of your conclusions.

Author Response

Q1. Lack of the line numbers.

A1. We added the line numbers in the revised manuscript.

Q2. What affects low coefficient of thermal expansion?

A2. We explained in detail that the key factors influencing the low CTE generation comprise the chemical factors (the main chain linearity/stiffness and the molecular weights) and physical factors (spontaneous in-plane orientation during solution casting), as described in the section 3.5 and Figure 14 in the revised manuscript.

Q3. TGA measurement: could you describe this method better? Range of temperatures etc?

A3. We added the conditions for the TGA measurements in the section 2.2.3 (yellow-marked sentences).

Q4. Optical transparency: what means the symbol in brackets?

A4. We corrected the garbled Greek character like “the wavelength (λ)” in the section 2.2.4 (yellow-marked)

Q5. The goal of the work should be presented in the Introduction section instead in the Results and Conclusions part.

A5. According to your advice, we moved the sentence regarding the goal of this work from the beginning of Results and Discussion to the end of Introduction (yellow-marked positions).

Q6. 3.4.1. The last sentence in this paragraph is not finished.

A6. We deleted the incomplete sentence “The previous report also”.

Q7. References: should be refreshed - most of them are too old. Also mostly Japanese authors are cited. It should be a balance with other authors who publish in this field as well, unless Japanese research dominates in this subject.

A7. We added 9 recent references (ref.11, ref.31-38) on optically transparent polyimides in the revised manuscript (yellow-marked).

Q8. Did you take photographs of these PIs?

A8. The photographs of the typical colorless PI films developed in this study are already shown in the bottom of Figure 14 in the initially submitted manuscript (rearranged at the bottom of Figure 15 in the revised manuscript)

Q9. Lack of explanation of your conclusions.

A9. We corrected the expressions and explanations in Conclusion and made it concise (yellow-marked).

Reviewer 3 Report

This paper presents a detailed study on synthesis and characterization of non-aromatic polyimides. The authors outline the challenges associated with the solubility and sought to find the processing conditions that lead to high molar masses and aligned backbone chains that can lead to low coefficient for thermal expansion. While the modified one-pot synthesis has been presented by the authors in previous publications, this paper showed that the combination of the cyclic aliphatic dianhydride with the rigid amide-containing diamines in the one-step polymerization yielded the target material. They hypothesized that the in-plane ordering of the main chain during the drying process allows for a low CTE film, along with other processing optimization for high performance polyimide materials.

The study is thorough, and the results are clearly displayed. I would be happy to see this manuscript being published if the authors address the following minor points.

  • Abstract is too long. The first half (up to the phrase “To solve this problem…”) is a long version of your motivation for this study, which is not warranted a fourteen-line description but rather a concise sentence or two. The introduction mostly repeats what is written here, and thus it is unnecessary to repeat.
  • Figure 12 alludes to the presence of liquid crystalline domains during the film casting process. Can this be visualized via in situ polarized optical microscopy? Also, the authors claim that the half-dried film with a Tg=Tdry is where the locally ordered domains start appearing. Do you have evidence for this?

Author Response

This paper presents a detailed study on synthesis and characterization of non-aromatic polyimides. The authors outline the challenges associated with the solubility and sought to find the processing conditions that lead to high molar masses and aligned backbone chains that can lead to low coefficient for thermal expansion. While the modified one-pot synthesis has been presented by the authors in previous publications, this paper showed that the combination of the cyclic aliphatic dianhydride with the rigid amide-containing diamines in the one-step polymerization yielded the target material. They hypothesized that the in-plane ordering of the main chain during the drying process allows for a low CTE film, along with other processing optimization for high performance polyimide materials.

The study is thorough, and the results are clearly displayed. I would be happy to see this manuscript being published if the authors address the following minor points.

Q1. Abstract is too long. The first half (up to the phrase “To solve this problem…”) is a long version of your motivation for this study, which is not warranted a fourteen-line description but rather a concise sentence or two. The introduction mostly repeats what is written here, and thus it is unnecessary to repeat.

A1. According to your advice, we made the sentences in Abstract concise as much as possible (yellow-marked).

Q2. Figure 12 alludes to the presence of liquid crystalline domains during the film casting process. Can this be visualized via in situ polarized optical microscopy? Also, the authors claim that the half-dried film with a Tg=Tdry is where the locally ordered domains start appearing. Do you have evidence for this?

A2. Thank you for your well-directed comments. This schematic illustration suggests the formation of a locally ordered structure during solution casting, in which the polymer chains can be highly aligned, as if they are liquid-crystalline domains. This hypothesis was inspired by our previous results: a half-imidized poly(amic acid) obtained by polyaddition of a imide-containing diamine, i.e., N,N-bis(4-anilino)-biphenyltetracarboxydiimide and s-BPDA can indeed form a lyotropic liquid crystal (LC) in NMP at a high solute content, and subsequent thermal imidization of the POM-detectable LC-remaining precursor cast film caused a prominent decrease in the CTE, compared with the counterpart from the LC-free PAA film prepared by the common route (SI 4, added in the revised version). The results suggest that LC-like ordered structures can contribute to reducing the CTE. However, in fact, no optical anisotropic textures were observed on polarizing microscope (POM) during the solution casting in our H-PMDA-based low-CTE systems. Thus, we do not have direct evidence of the presence of LC-like ordered structures in the present systems. However, if locally ordered regions had a much smaller size than the visible wavelength range (a detection limit of POM), our hypothesis can be reasonable.

  We added the related explanation and reference (#62) in the revised manuscript (p43, line17 - p44, line10, yellow-marked sentences). We also added a schematic diagram in Fig.13(b) in the revised version for better explanations.

Round 2

Reviewer 2 Report

Page 4
Line 2-3: Is so many citations necessary here?